# 3D Porous Structure-Inspired Lignocellulosic Biosorbent of *Medulla tetrapanacis* for Efficient Adsorption of Cationic Dyes

**DOI:** 10.3390/molecules27196228

**Published:** 2022-09-22

**Authors:** Jie Zhang, Hao Ji, Zepeng Liu, Liping Zhang, Zihao Wang, Ying Guan, Hui Gao

**Affiliations:** School of Forestry and Landscape Architecture, Anhui Agricultural University, Hefei 230036, China

**Keywords:** *Medulla tetrapanacis*, cationic dye, adsorption, biosorbent

## Abstract

The focus of this work was on developing a green, low-cost, and efficient biosorbent based on the biological structure and properties of MT and applying it to the remediation of cationic dyes in dye wastewater. The adsorption performance and mechanism of MT on methylene blue (MB) and crystal violet (CV) were investigated by batch adsorption experiments. The results demonstrated that the highest adsorption values of MT for MB (411 mg/g) and CV (553 mg/g) were greatly higher than the reported values of other biosorbents. In addition, the adsorption behaviors of methylene blue (MB) and crystal violet (CV) by MT were spontaneous exothermic reactions and closely followed the pseudo-second-order (PSO) kinetics and Langmuir isotherm. Further, the depleted MT was regenerated using pyrolysis mode to convert depleted MT into MT-biochar (MBC). The maximum adsorption of Cu^2+^ and Pb^2+^ by MBC was up to 320 mg/g and 840 mg/g, respectively. In conclusion, this work presented a new option for the adsorption of cationic dyes in wastewater and a new perspective for the treatment of depleted biosorbents.

## 1. Introduction

The treatment of dye wastewater has always been one of the challenges of water pollution. Cationic dyes have been employed in a wide range of fields, such as wool, printing and dyeing, and pharmaceuticals, etc. The presence of cationic dyes in wastewater has been reported to disrupt ecological balance [1] and posed certain risks to human health [2]. Therefore, wastewater would be purified to meet discharge standards. Various modalities have been developed to eliminate heavy metal ions and dye molecules from wastewater, including flotation [3,4,5], chemical precipitation, coagulation, bio-oxidation, ion exchange, membrane filtration, and adsorption, etc. [6]. The adsorption method has the characteristics of simplicity, high efficiency, and wide application, and is considered a feasible method for treating dye wastewater. Currently, various types of adsorbents have been designed to treat dye wastewater, including organic–inorganic composites [7], activated carbon [1], hydrogel [8], biochar [9], and other adsorption materials. However, adsorption materials are faced with problems such as unsatisfactory adsorption performance, high cost, and secondary pollution. Therefore, new adsorbents with high adsorption capacity, sustainability, green attributes, and non-toxicity are urgently needed to solve the problem of dye-polluted wastewater [10].

*Tetrapanax papyriferus* (*Araliaceae*) is mainly distributed in southwest China and is characterized by fast growth and high reproductive capacity. MT is a columnar, white, lightweight material derived from the pith of *Tetrapanax papyriferus*. The diameter and length of MT range as 5–30 mm and 80–100 cm, respectively. MT is very famous as traditional medicine in China, which is usually cut into slices for sale. The chemical composition of MT was determined in the previous study [11], and holocellulose reached 82.03%, lignin content was only 5.37%, and ash content and ethanol/benzene extractive content were 11.39% and 1.21%, respectively.

The primary chemical constituents of natural plants are cellulose, hemicellulose, lignin, and some minor components such as ethanol/benzene extractive, etc. The conversion of natural biomass into biosorbents with high adsorption properties has proven to be a promising research direction [12]. The feasibility of plant or industrial plant waste as potential biosorbents for cationic dyes has been demonstrated by different studies, such as with fallen leaves [13], *Pine elliottii* waste [14], walnut shells [15], straw and sugarcane waste [12], and tree bark [16]. These biosorbents have captured interest because of their extensive feedstock, environmentally friendly preparation processes, non-toxicity, and low cost. Compared to traditional biomass materials, MT possesses a higher holocellulose content and a special 3D porous microstructure. The high cellulose content means that the adsorbent surface has more electronegative oxygen-containing functional groups, which provide attachment sites for the adsorption of cationic dyes, while the regular porous structure provides wide diffusion channels for the dye molecules [17]. Therefore, MT is expected to be a biosorbent with excellent performance for cationic dyes. According to the discovery of Lawal et al. [18], the high content of cellulose contributed substantially to evolving mesopores, and biochar produced by high-cellulose biomass material was more effective in the treatment of dye wastewater. The study of Lawal provided a new idea for the treatment of depleted biosorbent. Additionally, citric acid was selected as a chemically modified reagent to treat MT. CA is an edible acid, which has been widely applied to modify various waste biomass materials to adsorb dyes. One of the advantages of this modification method is the use of non-toxic reactants [19]. A previous study reported that citric acid was used for kenaf fiber modification, and the adsorption efficiency for MB (131.6 mg/g) was significantly enhanced [20]. Similarly, pine sawdust modified with CA also improved the removal capability of MB (111.46 mg/g) [21]. It is known that related studies prepared biochar from MT as a substrate for supercapacitors [22] and as an adsorbent to adsorb heavy metal ions [11]. However, the excellent performance of MT itself has not been noticed. The studies related to the application of MT as a biosorbent for the removal of cationic dyes have not been reported.

In this study, the adsorption performance of MT for cationic dyes was evaluated using two typical cationic dyes (MB, CV) as model dyes. The adsorption mechanism of MT on cationic dyes was investigated. In addition, CA was applied as a modifier to improve the adsorption properties of MT. Finally, MBC was prepared using depleted biosorbents to avoid secondary pollution to the environment, and the adsorption performance of biochar on Cu^2+^, Pb^2+^, MB, and Congo red (CR) was detected.

## 2. Results and Discussion

### 2.1. Characterization of MT

MT was considered to be an excellent adsorbent material because of its well-developed pore structure, ultra-thin cell wall, and high holocellulose content. The SEM images of MT sections are shown in Figure 1a–c. As shown in Figure 1a, the cross-section of MT showed a regular honeycomb-like structure with an average pore size of about 200 µm. As can be seen in Figure 1b,c, both the radial and tangential sections of MT were observed to exhibit a honeycomb-like shape, which was obviously different from the other natural plant cell wall structures. Therefore, it could be inferred that the MT was a naturally occurring 3D penetrating porous structure. The 3D simulation of the MT spatial structure was shown in Figure 1f. As shown in Figure 1d,e, the MT pore walls became thicker and smoother after the adsorption of dyes. The phenomenon indicated that the originally white surface of MT was also changed to the corresponding color due to adsorption of dyes. These results demonstrated that MT had good adsorption capacity for cationic dyes. 

The FTIR spectra of MT (a), MT-MB (b), and MT-CV (c) are presented in Figure 2. The stretching vibrations of O-H and C-H bonds on biomass materials correspond to the peaks at 3340 cm^−1^ and 2920 cm^−1^, respectively. The peak at 1725 cm^−1^ was reported to be associated with -COO^-^ [23]. It was evident that the intensities of the absorption peaks of MT at 1335 cm^−1^ and 886 cm^−1^ were increased after the adsorption of MB. The bands at 1335 cm^−1^ and 886 cm^−1^ were attributed to the C-N bond in the MB molecules [24] and the C-H bond in the aromatic ring [25], respectively. A noticeable enhancement of the absorption peak at 1174 cm^−1^ was observed after the adsorption of CV, which was attributed to the C=N on the CV molecule [24]. In conclusion, the electronegative oxygen-containing functional groups on the surface of MT could be served as effective adsorption points for binding to cationic dyes.

The pore structure characteristics of different biosorbents and activated carbon are listed in Table 1. As seen in Table 1, the pore characteristics of MT were noticed to be similar to other biosorbents, but was significantly different from activated carbon. The specific surface area of MT (2.37 m^2^/g) was greatly lower than activated carbon (249 m^2^/g), indicating that the adsorption of MT on dyes was mainly identified to be related to the active groups on the surface of the biosorbent rather than physical filling [26]. The well-developed macropore structure of MT was considered to facilitate the swelling and promote the binding of dye molecules to the chemical groups on the MT surface [24].

### 2.2. Adsorption Study

#### 2.2.1. Effect of pH and MT Dosage on MB and CV Adsorption

The adsorption capacity of the biosorbent was significantly influenced by the pH of solution mainly in two aspects: (1) The degrees of ionization of solute molecules were controlled by the pH of the solution. (2) The pH value of the solution had an effect on the structural stability and surface chemical properties of biosorbent [29]. The adsorption mechanism of MT for cationic dyes can be further investigated by analyzing the influence of solution pH. According to the preliminary experiment, the adsorbance of the MB solution was observed to stabilize under acidic and slightly alkaline conditions (pH < 9), followed by a drastic decline with increasing pH of the solution (10 < pH < 13). According to Lin et al. [30], MB molecules had the property of converting into complex compounds under alkaline conditions. The solution of CV was observed to exhibit a blue color under acidic conditions (pH < 2) and gradually changed to green as the pH of the solution decreased. To exclude the interference of pH on the adsorbance of the dye solution, the initial pH of the dye solution was set to 3.0–9.0 in this study. The impact of pH on the adsorption capacity is presented in Figure 3a. According to Figure 3a, the adsorption capacity of MT for both MB and CV showed an increasing trend as the pH of the solution was increased. In an acidic medium, the surface of MT was rich in positive charge, and part of the active sites were occupied by protons. In an alkaline medium, the MT surface was deprotonated, and more active sites were exposed, which was responsible for the significant increase in MT adsorption capacity. Therefore, the initial pH of the dye solution was adapted to 9.0 for further experiments. Based on the above analysis, electron donor–acceptor may be an important mechanism for the adsorption of cationic dyes by MT, which was consistent with the study of Dallel et al. [31]. It was also reasonable to infer that MT loaded with cationic dyes could be desorbed and regenerated by competitive adsorption of H^+^ and MB^+^ under acidic conditions to achieve the purpose of recycling. This result had also been verified in subsequent experiments.

The influence of adsorbent dosage and removal rate is shown in Figure 3b. According to Figure 3b, the removal rate showed an increasing trend as the adsorbent dosage increased, which could be explained by the fact that more adsorbents provided more adsorption sites [32]. However, the number of dye molecules in a fixed volume of solution were limited. The excess of adsorption sites over the saturation adsorption requirement indicated that a lot of effective adsorption points were underutilized [33]. Obviously, the adsorption capacity of MT for MB and CV was significantly reduced when the amount of adsorbent addition was increased. Finally, the adsorbent dosage of 0.4 g/L was chosen to be applied in the following experiments.

#### 2.2.2. Effect of Adsorption Time on MB (CV) Adsorption and Kinetics Study

The influence of adsorption time on the adsorption capacity of MB and CV is displayed in Figure 4a. The adsorption equilibrium time of MT for MB was 960 min, and that of CV was 1500 min. The kinetic parameters of the MB and CV adsorption are presented in Figure 4b–d and Table 2. The PSO kinetic model had a better fit than the PFO dynamics model. Meanwhile, the adsorption capacity obtained by PSO kinetic fitting was closely in accordance with the experimental adsorption capacity. The adsorption data closely followed the PSO model, indicating that the adsorption for MB and CV by MT was dominated by chemisorption, including electronic exchange or sharing [34]. The fitting curves of the intra-particle diffusion model are presented in Figure 4c,d. The intra-particle diffusion model for MB and CV adsorption by MT could be divided into three stages, which was consistent with the study of Wang et al. [1]. In the first stage, the faster rate of adsorption was attributed to the large difference between the internal and external concentration of the adsorbent. In addition, the C value representing the boundary layer thickness tended toward 0, indicating that there was a favorable surface compatibility between the MB (or CV) molecules and MT. Therefore, intra-particle diffusion was identified as the main rate-controlling step. In the second stage, part of the efficient sites on the biosorbent were occupied, which led to a reduction in the adsorption rate. Adsorption to equilibrium was observed in the third stage, where the efficient sites of biosorbent were completely occupied, and the adsorption and desorption of dye molecules on MT reached a dynamic equilibrium.

#### 2.2.3. Effect of Initial Concentration on MB (CV) Adsorption and Isotherm Study

The influence of the initial concentration of the dye for the adsorption of MB and CV by MT was determined, as presented in Figure 5a. According to Figure 5a, the adsorption ability of MB and CV showed an increasing trend as the initial concentration of dye increased, and then tended toward equilibrium. This phenomenon ascribed to the presence of a concentration grading between the adsorbent and the bulk of the solution, and the larger molecular mass transfer force, was provided by the solution with higher initial concentration. When the concentration of dye further increased, the effective sites of the adsorbent were completely consumed, and the adsorption capacity tended to balance [35]. Adsorption isotherm is usually adopted to evaluate the complex interplay between adsorbent and adsorbate, which is another important method to explore the mechanism of adsorption [36]. Langmuir, Freundlich, and Temkin adsorption isotherms were employed in this study. According to Figure 5b,c and Table 3, the Langmuir isotherm provided high determination coefficients for the adsorption of MB (0.99) and CV (0.98). The maximum adsorption capacity (Q_max_) predicted by the Langmuir model was correlated closely with the experimental result, indicating that the adsorption process of MT for dye molecules was monolayer. Moreover, the maximum R_L_ values of MB (0.48) and CV (0.71) were calculated to be lower than 1, which was due to the adsorption procedure being favorable. The adsorption capacity of various materials for MB and CV are compared in Table 4. As evidenced by the table, MT has a good adsorption capacity for MB and CV.

#### 2.2.4. Thermodynamic Study

The thermodynamic parameters are shown in Figure 6 and Table 5. The spontaneous and exothermic characteristics of MT for MB and CV adsorption could be explained by the negative Gibbs free energy ($∆G^{o}$) and the negative enthalpy ($∆H^{o}$). This result indicated that the adsorption of MT for cationic dyes proceed spontaneously, which provided a good premise for the practical application of MT. Enthalpy change represented the energy that was changed during the adsorption process. When the adsorption process with lower energy change (absolute value of $\;∆H^{o}$ is 0–20 KJ/mol) was described as physical adsorption, and with higher energy change (absolute value of $∆H^{o}$ is 80–400 KJ/mol) it was described as chemical adsorption [44]. As presented in Table 5, the absolute values of the enthalpy of adsorption for MB and CV by MT were 25.75 and 14.98 KJ/mol, respectively. Analysis of the data showed that the adsorption of MT for cationic dyes was a complex procedure, which involved both physical and chemical adsorption. The negative entropy ($∆S^{o}$) for the adsorption of MB and CV by MT illustrated a promoted reordering with reduced randomness of biosorbent [10].

#### 2.2.5. Adsorption Mechanism

The adsorption of MT on cationic dyes was mainly influenced by the electronegative oxygen-containing functional groups (the hydroxyl and the carboxyl group) of cellulose and hemicellulose. The important role of oxygen-containing groups in the adsorption process was further validated by FTIR analysis. Cationic dyes can be immobilized by electronegative functional groups on MT through electron donor–acceptor. The adsorption mechanism was further discussed through the analysis of the adsorption model. The adsorption procedure closely followed the Langmuir isotherm and PSO kinetic model, suggesting that the adsorption procedure was monolayer adsorption and chemisorption, respectively. Meanwhile, intra-particle diffusion was the primary rate-controlling step in the adsorption of dye molecules by MT in the early stages of the adsorption procedure. It was noteworthy that the adsorption process may involve both chemical and physical processes by thermodynamic analysis, which suggested that hydrogen bonding and electrostatic interaction were equally important for the adsorption procedure [11]. The mechanism of MT adsorption of dyes is demonstrated by Figure 7. The unique 3D penetrating porous structure of MT was thought to facilitate swelling and provided channels for mass transfer of dye molecules.

#### 2.2.6. Reusability of MT

The regeneration performance is an important criterion for evaluating the quality of adsorbent. In this study, HCl (0.1 mol/L) was employed to remove the loaded dye molecules on MT. As seen in Figure 8, the adsorption capacity and removal rate of dyes by MT were observed to decrease slightly after three adsorption–desorption experiments, but MT still exhibited high adsorption capacity. However, the structure of the cell wall was destroyed by acids acting on glycosidic bonds hydrolyzing the matrix polysaccharides (cellulose and hemicellulose) [45]. In practice, the loose structure of MT after several cycles of desorption was not conducive to recovery. In this regard, MBC was prepared by pyrolysis of the depleted MT in nitrogen atmosphere. The adsorption properties of MBC for various contaminants are discussed in the next section.

#### 2.2.7. Study on Adsorption Capacity of MBC

The depleted adsorbent will cause secondary pollution to the environment if it is not properly treated. We innovatively recovered depleted MT, and the biochar was prepared under a high-temperature nitrogen atmosphere using the depleted MT as raw material. The results of exploration on the adsorption performance of MBC for MB, CR, Cu^2+^, and Pb^2+^ are presented in Figure 9. The adsorption capacity of MBC for Cu^2+^ and Pb^2+^ reached 320 mg/g and 840 mg/g, respectively, which was in accordance with the previous conclusion that MBC had good affinity for heavy metal ions. The adsorption capacity of MBC on CR and MB were 125 mg/g and 80 mg/g, respectively. Apparently, the adsorption performance of MBC on dyes did not meet our expectation. It will be our future work to improve the adsorption capacity of MBC on dyes.

#### 2.2.8. Performance Study of Citric Acid Modified MT

CA was employed as a modifier for the chemical modification of MT in this study. The modification effect of CA on MT was in two aspects. Firstly, the internal structure of MT was changed after the reaction with the CA solution. Secondly, the surface of MT introduced carboxyl groups through CA modification. The various properties of CAMT are presented in Figure 10. As evidenced by Figure 10a, the pore wall structure of MT was retained after CA modification, but the pore walls became thinner. It was noteworthy that inter-connected cellulose fiber networks were formed inside the CAMT. These fiber bundles were derived from the cell wall. During chemical treatment, part of the cellulose was detached from the cell wall and freeze-dried to form an interconnected network structure, which provided a better prerequisite for adsorption [46]. The change of the main peak of the MT and CAMT occurred at 1725 cm^−1^ in Figure 10b, which was caused by the introduction of carboxyl groups (COO-) after MT modification [47]. As seen in Figure 10c, the maximum adsorption of CAMT for MB enhanced from 400 mg/g to 526 mg/g, while the maximum adsorption of CV enhanced from 553 mg/g to 627 mg/g. After CA modification, the maximum adsorption capacities of CAMT for MB and CV were increased by 31.5% and 13.4%, respectively. From Figure 10d, it could be observed that the adsorption equilibrium times of MT for MB and CV were 960 min and 1500 min, and that of CAMT were 360 min and 720 min for MB and CV, respectively. The time consumed to achieve kinetic equilibrium was critical to evaluate the efficiency and feasibility of adsorbents for water pollution control. The adsorption equilibrium times of CAMT for MB and CV were reduced by 62.5% and 52%, respectively. The above data analysis demonstrated that CA modification can effectively reduce the time spent on the adsorption process and improve adsorption efficiency.

## 3. Materials and Methods

### 3.1. Materials

MT was obtained in Bozhou (Anhui province, China). Sodium bicarbonate (NaHCO_3_), copper nitrate (Cu(NO_3_)_2_), MB, sodium hydroxide (NaOH), CR, citric acid monohydrate (CA), CV, hydrochloric acid (HCl), and lead nitrate (Pb(NO_3_)_2_) were acquired from Xilong Scientific Co., Ltd. (Shantou, China). All other chemicals were used without further refinement. All experiments used deionized water.

### 3.2. Preparation of CAMT and MBC

MT was cut into the size of 5 × 5 mm and then placed in a beaker with 0.4 mol/L citric acid solution (*w*/*w* 1:50). The mixture was activated at 80 °C for 2 h, and then transferred to an autoclave lined with Teflon, and subsequently placed in an oven at 120 °C for 0.5 h for the esterification reaction. After completion of the reaction, MT was washed with NaHCO_3_ (0.1 mol/L) three times, and then deionized water was used to wash MT several times to remove residual NaHCO_3_. Finally, the modified biosorbent was freeze-dried, and named CAMT. The depleted biosorbent was dried at room temperature. The dried samples were placed in a vacuum high-temperature tubular sintering furnace (BTF-1200C-II, BEQ, Hefei, China) and pyrolyzed under nitrogen atmosphere. The heating rate was controlled at 5 °C/min, and the temperature was held at 700 °C for 2 h. After pyrolysis, the biochar samples were first washed with hydrochloric acid (10 wt %) to remove ash, and then with deionized water to neutral. After drying, the biochar was obtained and named MBC.

### 3.3. Characterization

Scanning electron microscope (VEGA3, TESCAN, Brno, Czech Republic) was applied to determine the physical structure of the samples. Before measurement, the samples were placed on conductive adhesive for vacuum gold spraying. Fourier-transform infrared spectrometer (Bruker Tensor II, Bruker, Karlsruhe, Germany) was applied to identify the functional groups of the samples. The pore structure of the samples was determined with the surface area and porosity analyzer (ASAP2460, Micromeritics, Norcross, GA, USA). The contents of MB and CV in aqueous solutions were measured at 665 nm and 590 nm by UV-vis spectrophotometer (TU-1810PC, PERSEE, Beijing, China), respectively. The concentrations of Cu^2+^ and Pb^2+^ in the samples were detected by atomic absorption spectrophotometer (TAS-990, PERSEE, Beijing, China).

### 3.4. Batch Adsorption Experiments

In batch adsorption experiments, 20 mg of biosorbent was added to a triangular flask containing 50 mL and 200 mg/L of MB (or CV) solution, the pH of the suspension was set to 9, and the suspension was subjected to a water bath oscillator at 303 K for 24 h at 120 rpm. The pH of the suspension was adjusted to 3.0–9.0 using 0.1 mol/L HCl or NaOH to explore the influence of pH on the adsorption capacity. The dosage of biosorbent was set at 10–50 mg to evaluate the influence of biosorbent dosage on the adsorption ability. The exploration process of MBC for MB, CR, Cu^2+^, and Pb^2+^ adsorption capacity was similar to the above procedure. All experiments were set up with three parallel groups of samples, and the relative error of the experimental data was within 5%. The calculation formulas of adsorption capacity and removal rate are as follows:(1)$$Q_{e}=\frac{\left(C_{0}-C_{e}\right)V}{m}$$
(2)$$Q_{t}=\frac{\left(C_{0}-C_{t}\right)V}{m}$$
(3)$$R=\frac{\left(C_{0}-C_{e}\right)}{C_{0}}\times 100\;\%$$
where Q_e_ and Q_t_ (mg/g) represent the adsorption capacities at adsorption equilibrium and at time t, respectively. C_0_, C_t_, and C_e_ (mg/L) represent the adsorbate concentrations in the solution at the initial time, the fixed time t, and adsorption equilibrium, respectively. V (L) represents the volume of solution. R (%) represents the removal rate. m (g) represents the weight of adsorbent used.

### 3.5. Adsorption Kinetics Study

The concentration of MB (or CV) in suspension at different time points within 24 h was determined to investigate the impact of adsorption time on MB (or CV) adsorption ability. To elaborate adsorption mechanism precisely, pseudo-first-order (PFO) model, PSO model, and intra-particle diffusion model are adopted to describe the adsorption process and the equations can be expressed as [19]:

the PFO model:(4)$$Q_{t}=Q_{e}\left(1-e^{-K_{1}t}\right)$$

the PSO model:(5)$$Q_{t}=\frac{Q_{e}{}^{2}K_{1}\times t}{Q_{e}K_{1}\times t+1}$$

the intra-particle diffusion model:(6)$$Q_{t}=K_{\mathrm{id}}t^{\frac{1}{2}}+C$$
where Q_e_ and Q_t_ (mg/g) represent the adsorption capacity at adsorption equilibrium and at time t, respectively. t (min) represents the adsorption time. C (mg/g) represents the intercept. K_1_ (min^−1^), K_2_ (g/mg × min), and K_id_ (g/mg × min) are the constant of the PFO rate, the PSO rate, and the intra-particle diffusion rate, respectively.

### 3.6. Adsorption ISOTHERM Study

The starting concentration of MB (or CV) was specified as 25–500 mg/L to evaluate the impact of the starting concentration on the adsorption ability. Adsorption isotherms are another critical methodology to study the adsorption mechanism and are crucial for the optimization and use of adsorbents [36]. Langmuir, Freundlich, and Temkin isotherm models were adopted to describe the equilibrium data. The equations are as follows [19]:

Langmuir:(7)$$Q_{e}=\frac{K_{L}C_{e}{\;Q}_{\max}}{1+K_{L}C_{e}}$$
(8)$$R_{L}=\frac{1}{1+K_{L}C_{0}}$$

Freundlich:(9)$$Q_{e}=K_{F}C_{e}{}^{\frac{1}{n}\;}$$

Temkin:(10)$$Q_{e}=\frac{\mathrm{RT}}{b_{T}}\ln K_{T}+\frac{\mathrm{RT}}{b_{T}}\ln C_{e}$$
where C_0_ and C_e_ (mg/L) represent the starting and at-adsorption equilibrium concentration of MB (CV), respectively. Q_e_ and Q_max_ (mg/g) represent the maximum adsorption value at equilibrium and theoretical maximum adsorption value, respectively. K_L_ (L/mg), K_F_ (mg × g^−1^ × mg^1/n^ × L^−1/n^), and K_T_ (L/mg) represent the Langmuir, Freundlich, and Temkin adsorption constant, respectively. n is the adsorption constant. R_L_ is a dimensionless separation coefficient for determining whether the adsorption process is favorable. T (K) is the absolute temperature. b_T_ (KJ/mol) represents the characteristic constant of the Temkin model. R (8.314 J/mol × K) represents the universal gas constant.

### 3.7. Adsorption Thermodynamic Study

The temperature range was set to 303–323 K to explore the influence of different temperatures on the adsorption ability of MT. The adsorption mechanism and the driving force of adsorption can be further explained through the analysis of adsorption thermodynamics. Gibbs free energy ($∆G^{o}$, KJ/mol), enthalpy ($∆H^{o}$, KJ/mol), and entropy ($∆S^{o}$, J/mol·K) are calculated as follows [6]:(11)$$K_{d}=\frac{Q_{e}}{C_{e}}$$
(12)$$\ln K_{d}=\frac{∆S^{o}}{R}-\frac{∆H^{o}}{\mathrm{RT}}$$
(13)$$∆G^{o}=∆H^{o}-T∆S^{o}$$
where Q_e_ (mg/g) represents the adsorption capacity. C_e_ (mg/L) represents the MB (or CV) concentrations. T (K) represents the temperature in Kelvin. R (8.314 J/mol × K) represents the universal gas constant. K_d_ is the thermodynamic equilibrium constant.

### 3.8. Regeneration Experiment

Desorption cycle experiment was used to assess the reusability of MT. An amount of 20 mg MT was added to a conical flask with 50 mL MB (or CV) solution (150 mg/L). The MT was collected after equilibrium adsorption and desorbed in HCl solution (0.1 mol/L) for 24 h. After desorption, MT was washed to neutral and then freeze-dried. The cycle experiment was repeated 3 times, and the adsorption capacity of MT in each cycle was measured.

## 4. Conclusions

In this study, MB and CV were used as model dyes, and the adsorption potential of MT for cationic dyes was evaluated. Further, the mechanism of MT adsorption for cationic dyes was investigated. The main findings are summarized as follows:

(1) The maximum adsorption capacities of MT for MB and CV reached 411 mg/g and 553 mg/g, respectively. In addition, the adsorption efficiency of MT for dyes could be effectively improved by CA modification. Meanwhile, MT still possessed good adsorption performance after three adsorption–desorption experiments. The above data proved that MT was a promising biosorbent for the treatment of cationic dye contamination.

(2) The thermodynamic study demonstrated that the adsorption process was spontaneous and exothermic. The adsorption procedure closely followed PSO kinetic, and the intraparticle diffusion was the main rate-controlling step. The 3D pore structure of MT was mainly used for mass transfer of dye molecules rather than pore filling.

(3) The maximum adsorption capacities of MBC for Cu^2+^ and Pb^2+^ were 320 mg/g and 840 mg/g, respectively, which indicated that the preparation of biochar for adsorption of heavy metal ions using depleted MT as a carbon source is a feasible method to treat depleted MT.

## Figures and Tables

**Figure 1 molecules-27-06228-f001:**
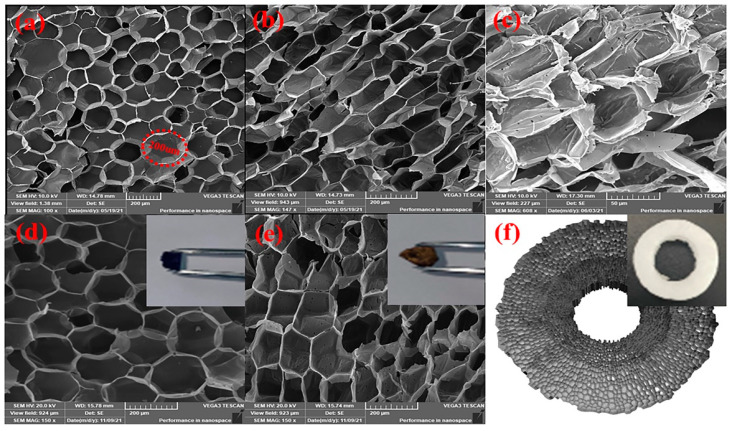
SEM images of MT (**a**) cross-section, (**b**) radial-section, and (**c**) tangential-section. SEM images of MT cross-section after adsorption of (**d**) MB, (**e**) CV. (**f**) The 3D simulation of the MT spatial structure.

**Figure 2 molecules-27-06228-f002:**
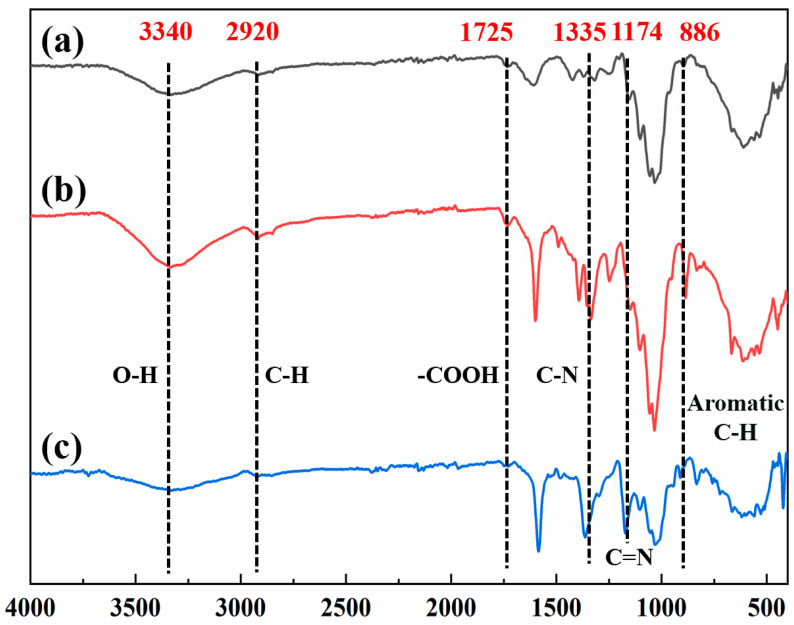
FTIR spectra of (**a**) MT, (**b**) MT-MB, and (**c**) MT-CV.

**Figure 3 molecules-27-06228-f003:**
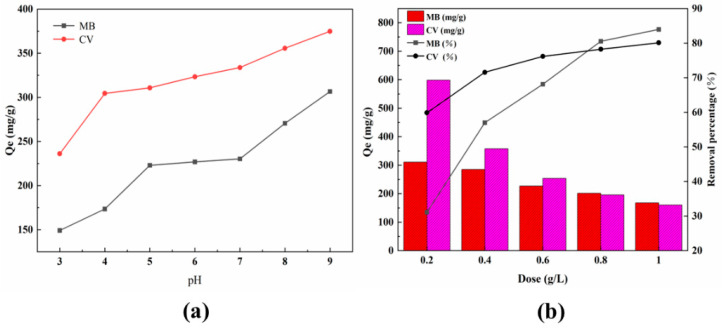
The effect of (**a**) pH and (**b**) biosorbent dosage on MT adsorption capacity.

**Figure 4 molecules-27-06228-f004:**
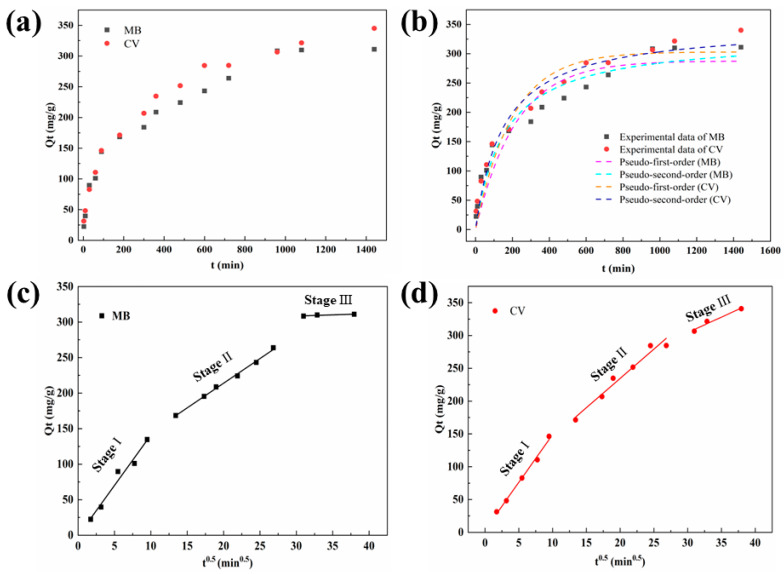
(**a**) The effect of contact time on the performance for the adsorption of MB and CV by MT. Adsorption kinetics for the adsorption of MB and CV by MT: (**b**) pseudo-first-order model, pseudo-second-order model, (**c**,**d**) intra-particle diffusion model.

**Figure 5 molecules-27-06228-f005:**
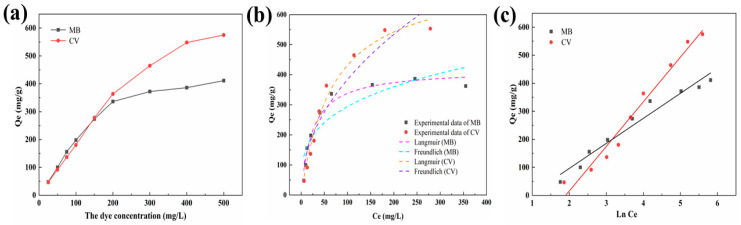
(**a**) The effect of initial concentration on the performance for the adsorption of MB and CV by MT. Adsorption isotherms for the adsorption of MB and CV by MT: (**b**) Langmuir model, Freundlich model, and (**c**) Temkin model.

**Figure 6 molecules-27-06228-f006:**
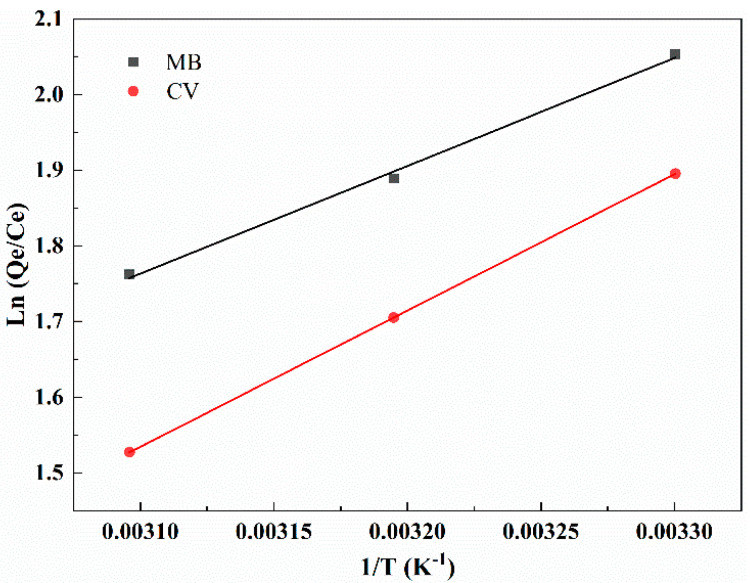
The van’t Hoff plot for the adsorption of MB and CV by MT.

**Figure 7 molecules-27-06228-f007:**
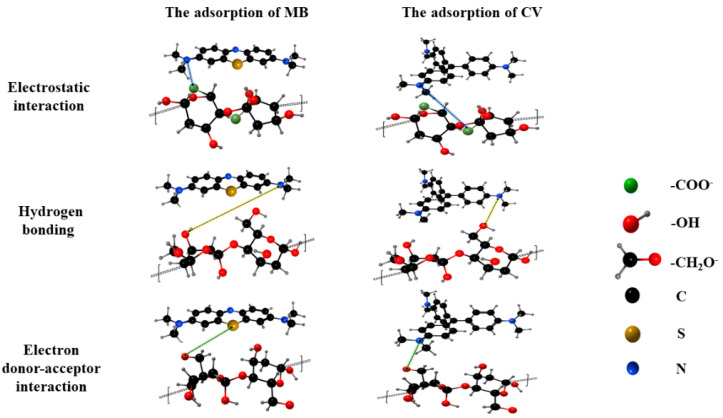
Possible adsorption mechanism.

**Figure 8 molecules-27-06228-f008:**
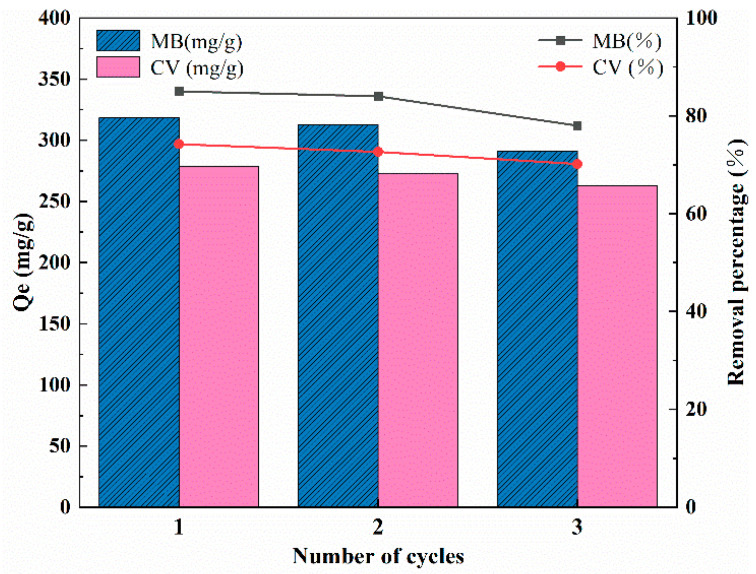
Reusability of MT for the adsorption of MB and CV.

**Figure 9 molecules-27-06228-f009:**
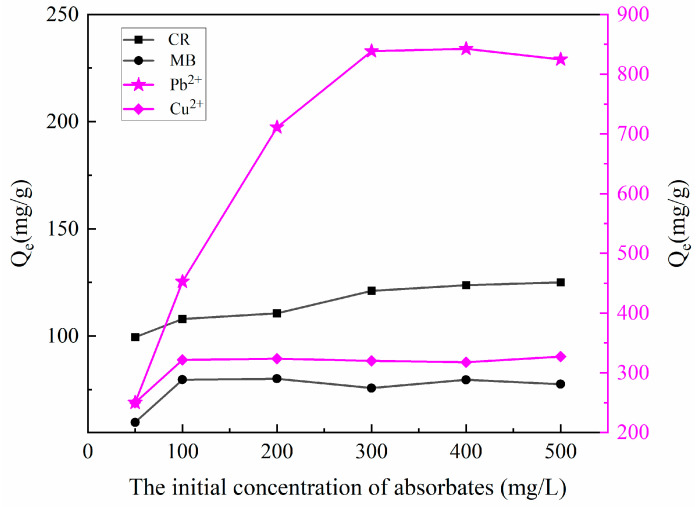
The adsorption capacity of MBC for Cu^2+^, Pb^2+^, MB, and CR.

**Figure 10 molecules-27-06228-f010:**
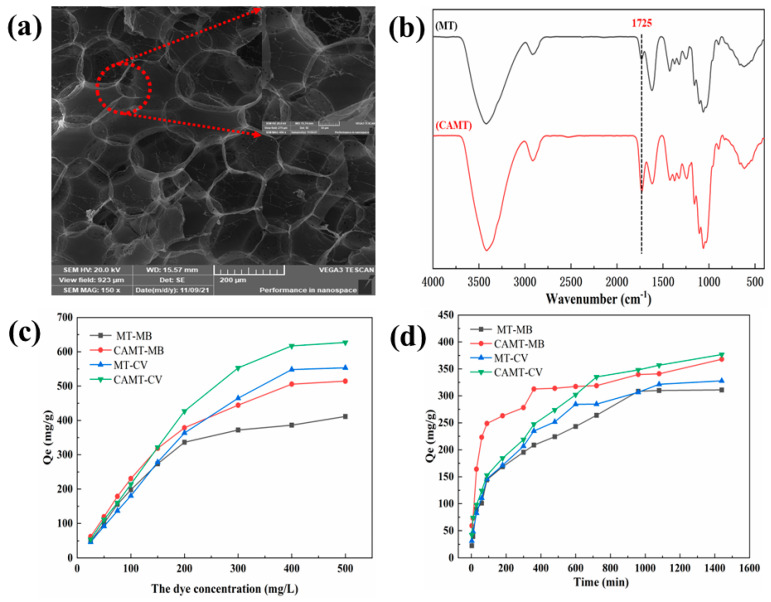
(**a**) SEM image of CAMT. (**b**) FTIR spectra of MT and CAMT. (**c**) Effect of initial concentration of dye on the performance of MT and CAMT. (**d**) Effect of adsorption time on the performance of MT and CAMT.

**Table 1 molecules-27-06228-t001:** Pore characteristics of different adsorbents.

Adsorbents	Surface Area (m^2^/g)	Pore Volume (m^3^/g)	Pore Diameter (nm)	Sources
Straw	0.37	--	--	[26]
Eucalyptus leaves	7.40	0.0086	--	[27]
Leaves of Magnoliaceae	7.85	0.0088	7.05	[13]
Sunflower pith	64.49	0.0500	1.34	[23]
Activated carbon	249	0.1330	3.09	[28]
MT	2.37	0.0026	4.41	This study

**Table 2 molecules-27-06228-t002:** Adsorption kinetics and intra-particle diffusion model parameters for the adsorption of MB and CV by MT.

Model	Parameters	MB	CV
Pseudo-first-order	Q_e_ (mg/g)	287.70	303.09
	K_1_ (min^−1^)	0.0046	0.0050
	R^2^	0.89	0.94
Pseudo-second-order	Q_e_ (mg/g)	328.99	348.97
	K_2_ (g/mg·min)	0.000019	0.000018
	R^2^	0.95	0.97
Intra-particle diffusion	K_id1_ (mg·g^−1^·min^−0.5^)	14.19	14.52
	C_1_ (mg/g)	0.91	3.54
	R^2^	0.97	0.99
	K_id2_ (mg·g^−1^·min^−0.5^)	7.20	8.93
	C_2_ (mg/g)	69.23	53.96
	R^2^	0.99	0.96
	K_id3_ (mg·g^−1^·min^−0.5^)	0.34	2.67
	C_3_ (mg/g)	298.12	227.92
	R^2^	0.89	0.78

**Table 3 molecules-27-06228-t003:** Adsorption isotherm parameters for the adsorption of MB and CV by MT.

Dyes	Langmuir				Freundlich			Temkin		
	Q_max_ (mg/g)	K_L_ (L/mg)	R^2^	R_L_	K_F_ (mg·g^−1^·mg^1/n^·L^−1/n^)	n	R^2^	K_T_ (L/g)	b_T_ (J/mol)	R^2^
MB	400	0.043	0.99	0.044–0.48	38.28	2.2	0.83	0.40	28.23	0.96
CV	588	0.016	0.98	0.11–0.71	16.98	1.5	0.94	0.15	15.83	0.97

**Table 4 molecules-27-06228-t004:** Comparison of adsorption capacity of different adsorbents for MB and CV.

Adsorbents	Adsorption Capacity (mg/g)	Dyes	Sources
CMC/PAA/GO composite	138.4	MB	[37]
Rapanea ferruginea	106	CV	[38]
Tea waste/Fe_3_O_4_ magnetic composite	333.3	CV	[39]
Hickory chip biochars	310	MB	[40]
PVDF/PDA/PPy composite	370.4	MB	[41]
CNC/MnO_2_/SA composite	114.5	MB	[8]
Biochar from crisp persimmon peel	59.7	MB	[42]
ZnO-Chitosan Nanocomposites	97.9	MB	[43]
MT	400	MB	This study
MT	553	CV	This study

**Table 5 molecules-27-06228-t005:** Adsorption thermodynamic parameters for the adsorption of MB and CV by MT.

Dyes	$∆S^{o}$ (J/mol × K)	$∆H^{o}\;(\mathrm{KJ}/\mathrm{mol})$	$∆G^{o}\;(\mathrm{kJ}/\mathrm{mol})$			R^2^
			303K	313K	323K	
MB	−72.50	−25.75	−3.79	−3.06	−2.34	0.99
CV	−33.68	−14.98	−4.77	−4.43	−4.10	0.99

## Data Availability

The data supporting this study are available when reasonably requested from the corresponding author.

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
