# Peer review of "3D Porous Structure-Inspired Lignocellulosic Biosorbent of Medulla tetrapanacis for Efficient Adsorption of Cationic Dyes"

_molecules, 2022, doi:10.3390/molecules27196228_

Round 1
Reviewer 1 Report
Comments from Reviewer
Title: 3D porous structure-inspired lignocellulosic biosorbent of Medulla tetrapanacis for efficient adsorption of cationic dyes
The current form's presentation of methods and scientific results is satisfactory for publication in the Molecules journal. The minor and significant drawbacks to be addressed can be specified as follows:
1. Line 79. biochar (MBC) ---> MT-biochar (MBC).
2. Some abbreviations are explained several times in the text. It is sufficient to do it the only first time, for example, Lines 79 and 366.
3. Lines 139, 141, and 143. The ---> the.
4. Tab. 1. sources ---> Sources.
5. Fig. 3. On panel b Qe reaches the values of Qe(mg/g) 600 (CV) and 300 (MB) - these values are not visible on panel a.
6. Poor quality of the figures. Incompatible ranges ((i) upto 1500 or 1600 (ii) Qt uptov350 or 375). Different font sizes ((i) title: t^0.5… (ii) Qt (subscript). Panel a - there is no need to connect the points.
7. Lines 306 and 307. correlation coefficients ---> determination coefficients. R - correlation coefficients. R^2 - determination coefficients.
8. Tab. 4. adsorption capacity ---> Adsorption Capacity.
9. 3.2.5. Adsorption mechanism. What is new about the mechanism discussed in this paragraph compared to that discussed in the literature?
Sincerely,
The reviewer.
Reviewer 2 Report
In this paper, the adsorption performance of MT for cationic dyes was evaluated using two typical cationic dyes (MB, CV) as model dyes. The adsorption mechanism of MT on cationic dyes was investigated. In addition, CA was applied as a modifier to improve the adsorption properties of MT. It is an interesting content, but arranged structure needs to be further improved. Therefore, it needs minor revision before it is published in this journal. Some issues should be carefully addressed. Please check the attachment.

Round 2
Reviewer 1 Report
Congratulations on a great job. The author has made a substantial improvement for this article. The manuscript can be accepted for publishment in the present form.